# Novel Polyclonal Antibody Raised against Tetrodotoxin Using Its Haptenic Antigen Prepared from 4,9-anhydrotetrodotoxin Reacted with 1,2-Ethaneditiol and Further Reacted with Keyhole Limpet Hemocyanin

**DOI:** 10.3390/toxins11100551

**Published:** 2019-09-20

**Authors:** Shigeru Sato, Suzuka Takaishi, Ko Yasumoto, Shugo Watabe

**Affiliations:** School of Marine Biosciences, Kitasato University, Sagamihara-shi, Kanagawa 252-0373, Japan; mf16004@st.kitasato-u.ac.jp (S.T.); yasumoto@kitasato-u.ac.jp (K.Y.); swatabe@kitasato-u.ac.jp (S.W.)

**Keywords:** tetrodotoxin, tetrodotoxin analogs, pufferfish, polyclonal antibody, ELISA

## Abstract

A novel polyclonal antibody against tetrodotoxin (TTX) was raised using its haptenic antigen, where 4,9-anhydroTTX was reacted with 1,2-ethanedithiol and this derivative was further reacted with keyhole limpet hemocyanin (KLH). This newly designed antigen (KLH-TTX) was inoculated into rabbits, resulting in the production of the specific polyclonal antibody, which reacted well with TTX and its analogs, 4-*epi*TTX, 11-oxoTTX and 5,6,11-trideoxyTTX, except for 4,9-anhydroTTX. The enzyme-linked immunosorbent assay (ELISA) system using this specific antibody was also developed in the present study. This newly developed polyclonal antibody with analytical procedures using direct one-step ELISA is useful to detect TTX and its analogs in toxic organisms and also disclose the mechanisms involved in their metabolic pathways and accumulation of TTX.

## 1. Introduction

Tetrodotoxin (TTX), one of the most potent natural neurotoxins [1,2], has been first detected in pufferfish [3,4] and later determined for its unique structure by Tsuda et al. [5], Woodward [6] and Goto et al. [7]. Subsequently, TTX was found in various organisms, including vertebrates and invertebrates [8,9,10,11] and even bacteria [12,13,14]. These findings suggest that TTX is first produced in microbes and then accumulated in invertebrate and vertebrates through food webs [15,16]. However, bacteria produce TTX very marginally and the mechanisms involved in the accumulation of TTX at large amounts in eukaryotes have remained unknown [16,17]. Recent studies revealed that various TTX analogs, such as 4-*epi*TTX, 11-oxoTTX and 5,6,11-trideoxyTTX, exist together with TTX in diverse animals [18], although their concentrations in toxic organisms are much lower than those of TTX except 5,6,11-trideoxyTTX [19,20,21,22,23,24]. Their metabolic pathways to be converted to TTX have remained largely unknown. 

To cope with such uncertainties, it is important to develop sensitive methods to detect TTX or its analogs specifically. Such approaches to certainly determine small amounts of these compounds will help find metabolic pathways and biological significance of TTX in various organisms. One of such approaches is to obtain a specific antibody. However, a monoclonal antibody so far prepared reacts with only TTX, but hardly with its analogs [25,26,27,28]. Recently developed liquid chromatography connected with mass spectrometry has very high sensitivity to the above-mentioned compounds with certain specificity [20,21]. However, this approach has some limitations where TTX and its analogs cannot be well discriminated from contaminants when biological materials or their partially purified samples are directly analyzed. These contaminants are usually present in the extracts from TTX-containing organisms [20,21]. If we can raise a polyclonal antibody specific to TTX and its analogs, we can detect these compounds even at a marginal amount and even in the presence of contaminants. Such sensitive and multipotent antibody is expected to make clear the above-mentioned ambiguities, for example, localization of TTX in TTX-containing animals [29,30,31,32,33,34] and possibly even TTX analogs or metabolites without TTX. Such antibody is also desired in sanitary backgrounds to avoid any happenings to ingest toxic animals [30,31].

The present study developed a novel polyclonal antibody against TTX using its haptenic antigen, where 4,9-anhydroTTX was reacted with a dithiol reagent and this derivative was further reacted with keyhole limpet hemocyanin (KLH). This procedure has been successfully adopted to raise a polyclonal antibody against paralytic shellfish toxin (PSTs) [35,36]. The antibody in the present study reacted with not only TTX but also its analogs, although the reactivity of the antibody varied depending on the toxin structures. The enzyme-linked immunosorbent assay (ELISA) system using this novel antibody was also developed in this study.

## 2. Results

### 2.1. Reaction of 4,9-anhydroTTX and Dithiol Reagents

Yotsu-Yamashita et al. [20] reported that L-cysteine reacts with 4,9-anhydroTTX and forms an adduct in which C4 position of TTX couples with the sulfur atom of L-cysteine. In the present study, 4,9-anhydroTTX (initial concentration: 1.0 mM) in the reaction mixture with 1,2-ethaneditiol (EDT; Sigma-Aldrich, St. Louis, MO, USA) decreased during incubation at 37 °C and completely disappeared after 30 min. Similar results were observed when 4,9-anhydroTTX was incubated with (±)-dithiothreitol (DTT). After incubation for 30 min, the reaction mixtures with EDT or DTT showed no peak corresponding to 4,9-anhydroTTX, 4-*epi*TTX nor TTX in the HPLC fluorometric analysis (Appendix A). These results suggest that 4,9-anhydroTTX reacted with EDT and DTT to form adducts, in which their dithiols coupled to the C4 position of TTX. 

### 2.2. KLH-TTX Conjugate Used for Antigen

Figure 1 shows the estimation structure of antigen (KLH-TTX conjugate). A 21.6 µg amount of antigen was obtained as a precipitate after dialysis. When a 216-µg equivalent portion was suspended in 1 mL of water, a peak corresponding to 36.6 µM of TTX was detected in the fluorometric analysis, where KLH itself showed no fluorescence. This result indicates at least 5.4% (*w/w*) of TTX was coupled with KLH.

### 2.3. Development of Antibody against TTX

High titers appeared after immunization for one month. In Figure 2, titers are expressed as the amount of TTX (nmol) adsorbed in 1 mL serum. One rabbit (No.1) died during immunization, and four rabbits survived until the final stage. Titers at the first stage (pre-inoculation), corresponding to non-specific adsorption, were less than 1.0 nmol/mL serum, whereas those at the final stage (14th) varied from 4.0 to 24.5 nmol/mL serum. The antisera (No.2–No.5 rabbits, 7th) also adsorbed 4-*epi*TTX, 11-oxoTTX and 5,6,11-trideoxyTTX at 60 to 80 mol% amount of TTX, but hardly adsorbed 4,9-anhydroTTX (less than 1.0 nmol/mL serum). The amounts of TTX and its analogs trapped in No.2 rabbit are shown in Figure 3.

### 2.4. Conjugate of Biotin and TTX

The conjugate of biotin and TTX (Biotin-TTX, Figure 4) was prepared by the reaction of DTT-TTX with maleimide-PEG2-biotin. The conjugate in the reaction mixture was purified by Bio-Gel P-2 column chromatography and 1.8 µg amount of biotin-TTX (1.73 µmol) was obtained as a white powder. A part of the conjugate thus obtained was used to analyze its conformation by TOFMS. The data showed significant peaks at m/z 491.1707 and m/z 981.3337, corresponding to the pseudo-molecular ion peaks of m/z [M + 2H]^++^ (m/z calcd. 491.1723) and [M + H]^+^ (*m*/*z* calcd. 981.3368) of the target compound, respectively.

### 2.5. Reactivity of the Antibody to TTX and Its Analogs and PSTs in ELISA

Direct one-step ELISA was carried out using the purified antibody obtained from rabbit No.3. Biotin-TTX described above was used as a labeled toxin. As shown in Figure 5, detection limit and working range for TTX were around 3 nM and 10–300 nM, respectively. The association constant for TTX was 3.42 × 10^8^ M^−1^, whereas the range for quantification was between 10 and 100 nM. TTX analogs, such as 4-*epi*TTX, 11-oxoTTX, 5,6,11-trideoxyTTX, showed high reactivity to the antibody in ELISA, whereas 4,9-anhydroTTX did not. TTX and its analogs (TTXs) are known to block voltage-dependent sodium channel specifically [1,2]. The chemical structures of paralytic shellfish toxins (PSTs) are very different from those of TTXs, though PSTs have the same pharmacological activity. As shown in Figure 6, no cross-reaction was observed when PST components, GTX1/4, GTX2/3, C1/2, neoSTX nor dcSTX, were separately analyzed by the ELISA. Thus, the antibody obtained in the present study distinguishes between TTXs and PSTs. In addition, KLH itself did not react with the antibody (Appendix A). 

## 3. Discussion

A newly designed antigen (KLH-TTX) was inoculated into five rabbits for seven months to obtain a specific polyclonal antibody, which reacted well with TTX and its analogs except for 4,9-anhydroTTX. Because 4-*epi*TTX, 11-oxoTTX and 5,6,11-trideoxyTTX showed high cross-reactivity, C9 position of TTXs seems to be an important part for the recognition by the antibody.

Not only TTX but also various TTX analogs have been found in toxic organisms such as pufferfish [18,19]. Among these analogs, 5,6,11-trideoxyTTX could be a de novo product of TTXs (Figure 7), although biological conversions from deoxy derivatives to TTX have not been clearly demonstrated along with the origin(s) of TTX. Yasumoto et al. [11] reported that some marine bacteria produce TTX and 4,9-anhydroTTX, but the amount produced by these bacteria cannot well explain a high level of TTX accumulated in toxic organisms. TTX itself is water-soluble and likely to be hardly accumulated in organisms through the food web. It has been reported that the monoclonal antibody developed for TTX by Kawatsu et al. has quite low cross-reactivity for 4,9-anhydroTTX [25]. As shown in Figure 3 and Figure 5, the antibodies obtained from rabbits No. 2 and No. 3 also hardly recognized 4,9-anhydroTTX. Actually, the antiserum from rabbit No. 3 showed the lowest titer as shown in Figure 2. However, subsequent experiments for purification of the antiserum were performed with this rabbit antiserum, because we aimed to save the antisera of other rabbits for future investigations such as cellular localization of TTX and its analogs and examination of their metabolic pathways.

Development of new antibody for this toxin is expected to be produced in the future. Cross-reactivities of the other TTX derivatives such as 5,6,11-trideoxyTTX and 11-oxoTTX by the monoclonal antibodies already developed for TTX have not been reported. This is the first report of a new antibody which has wide range affinities on various TTX analogs.

The ELISA system using this specific antibody was also developed in the present study. So far, pufferfish toxin, mainly TTX, has been quantified by mouse bioassay [37]. However, mouse bioassay has negative aspects because it requires the use of laboratory animals and its sensitivity is not efficient. Recently, methods of liquid chromatography with MS detector as well as post-column derivatization fluorometric monitor have been developed to detect TTX and its analogs [21,38]. Currently, however, only a limited number of authorized references of TTX analogs are available for quantification of these LC analyses. Immuno-chemical methods have high sensitivity and specificity, and usually, a number of samples can be treated at a time. We are sure that our newly developed polyclonal antibody with analytical procedures using direct one-step ELISA is useful, not only to detect TTX and its analogs in various aquatic organisms, but also to disclose the mechanisms involved in metabolic pathways and accumulation of TTXs in toxic organisms. 

No cross-reaction was observed when PST components such as GTX1/4, GTX2/3, C1/2, neoSTX nor dcSTX were analyzed by the ELISA. TTXs and PSPs are voltage-dependent sodium channel blockers. The guanidino groups of TTXs and PSPs are essential to this pharmacological activity. The antibody in the present study is obtained from a novel antigen in which C4 position of TTX is coupled to a carrier protein while the guanidino group of TTX remains. The structure of antigen (Figure 1) can explain that the specific structures of TTX and its analogs except 4,9-anhydroTTX were well recognized by the polyclonal antibody, but not for PSTs. 

ELISA technique for summed PSTs concentration has been reported [36,39]. In comparison to them, sensitivity for TTX on ELISA in the present study seems to be insufficient. Some improvements may be achieved when the antisera of rabbits No. 2 and 5 is used after purification. 

## 4. Conclusions

A polyclonal antibody against TTX based on a new haptenic antigen was developed. This antibody reacts well with not only TTX but also 4-*epi*TTX, 11-oxoTTX, 5,6,11-trideoxTTX but not with 4,9-anhydroTTX, indicating the deference on C9 position of TTX analogs affects significantly on the cross-reaction. A direct one-step ELISA system using this novel antibody well recognizes various TTX analogs in high sensitivities. The polyclonal antibody newly designed ELISA system can be expected as useful tools for studies on distributions of TTX and its analogs, as well as biosynthetic mechanisms of TTX in toxic aquatic organisms. 

## 5. Material and Methods

### 5.1. Materials

Toxic pufferfish *Takifugu pardalis*, *T. snyderi*, *T. flavipterius* (former *T. poecilonotus*), and *T. porphyreus* were collected from a fish market in Ofunato, Iwate Prefecture, Japan, from April to July in 2016. Frozen samples of *Takifugu rubripes* and *Lagocephalus sceleratus* were supplied from the Enoshima Aquarium, Fujisawa, Kanagawa Prefecture, Japan, in December 2016. 

### 5.2. Preparation of TTX and Its Analogs

TTX and its analogs were extracted from the visceral part of fish with 0.1% acetic acid by heating in a boiling water bath for 10 min, according to the method for TTX described in Standard Methods of Analysis in Food Safety, Japan [37]. TTX and its analogs, 4,9-anhydroTTX and 5,6,11-trideoxyTTX, were isolated by successive column chromatography on activated charcoal (for chromatography; Wako, Osaka, Japan), Bio-Gel P-2 (200–400 mesh; Bio-Rad, Hercules, CA, USA), and Bio-Rex 70 (200–400 mesh; Bio-Rad, Hercules, CA, USA) according to the method described by Nagashima et al., [14]. 4,9-anhydroTTX thus isolated was used to prepare the antigen. In addition, an aliquot of isolated TTX (ca. 10 µmol) was dissolved in 5 mL of 0.1% acetic acid and heated in boiling water for 30 min. 4-*epi*TTX and 4,9-anhydroTTX produced in the reaction mixture were isolated by Bio-Rex 70 column chromatography (200–400 mesh, 1.5 × 115 cm) and used as the materials for the examination of their cross-reactions with the polyclonal antibody against TTX, which was produced in the present study as described below. Moreover, 11-oxoTTX was prepared from the isolated TTX by treating with the FeSO_4_ and H_2_O_2_ mixture, according to Wu et al. [40] (Appendix A).

### 5.3. Preparation of the EDT and TTX Adduct

Referring to the reaction between 4,9-anhydroTTX and L-cysteine [18], the adduct of TTX and EDT was prepared as follows. Thirty µmol of 4,9-anhydroTTX isolated from the pufferfish was dissolved in 20 mL of 0.05 M ammonium phosphate buffer (pH 8.0) and mixed with 10 mL of dimethylsulfoxide (DMSO) containing 260 µL of EDT. The mixture was incubated at 37 °C for 30 min, and the produced EDT-TTX adduct was extracted four times with an equal volume of ethyl acetate to remove remaining EDT and DMSO. The extract was loaded on a column of Bio-Gel P-2 (200–400 mesh, 1.5 × 10 cm) to purify the EDT-TTX adduct. The fractions containing the EDT-TTX adduct eluted in diluted acetic acid were combined, lyophilized and dissolved in 1 mL of 0.1 M sodium phosphate buffer (pH 7.2). Meanwhile, 20 mg keyhole limpet hemocyanin-HG (KLH; CAS. No. 9013-72-3; Wako, Japan) dissolved in 2 mL of 0.1 M sodium phosphate buffer (pH 7.4) was mixed with 10 mg *N*-(4-maleimidobutyryloxy) succinimide (GMBS, CAS. No. 80307-12-6; Dojindo, Masuki, Japan) dissolved in 0.3 mL of N-dimethylformamide. The mixture was allowed to stand at room temperature for 30 min, applied to a column of Sephadex G-25 (fine, 1.5 × 35 cm; GE Healthcare, Chicago, IL, USA), and eluted with 0.1 M sodium phosphate buffer (pH 6.0). While monitoring the eluate by absorption at 280 nm, fractions containing the KLH-GMBS conjugate were collected and adjusted pH to 7.2 by the addition of 0.1 M sodium hydroxide. The KLH-GMBS conjugate thus obtained was mixed with the EDT-TTX adduct prepared as described above. The mixture was allowed to stand at 5 °C overnight, dialyzed twice against 1 L of 0.1% acetic acid, and then twice against 1 L of phosphate-buffered saline without Ca^++^ and Mg^++^ [PBS(−)], resulting in the production of the TTX-KLH conjugate. 

### 5.4. Analytical Procedures

#### 5.4.1. Confirmation of TTX, 4-epiTTX, and 4,9-anhydroTTX by HPLC-FLD

TTX, 4-*epi*TTX and 4,9-anhydroTTX were quantified by fluorometric HPLC using post-column derivatization developed by Yotsu et al. [38] with minor modifications. The details of the HPLC fluorometric analysis are as follows: HPLC column, J-Pak Symphonia C18 (5 µm, 4.6 × 150 mm; Jasco, Tokyo, Japan); mobile phase, 0.06 M heptafluorobutiric acid (HFBA, 98%; Sigma-Aldrich, St. Louis, MO, USA) /0.05 M ammonium acetate buffer (pH 5.0), 0.4 mL/min; reaction reagent, 4 M sodium hydroxide, 0.4 mL/min; detector, FP-2020 Plus (Jasco, Ex 365 nm, Em 510 nm, Gain ×1000); pump for mobile phase, PU-2080 Plus (Jasco, Tokyo, Japan); pump for reaction reagent, PU-2080 Plus (Jasco, Tokyo, Japan); reaction coil, i.d.0.5 mm × 200 cm (stainless, 120 °C in dry oven); integrator, Chromatocorder 21 (SIC, Tokyo, Japan); injection volume, 10 µL (overload injection).

#### 5.4.2. EDT-TTX Adduct and TTX Coupled to KLH

The amounts of EDT-TTX and TTX coupled to KLH were estimated by the same system as shown in Section 5.4.1. without an HPLC column. Fluorometric peak intensity was compared with that of TTX standard solution (10 µM), and its concentration was expressed as TTX equivalent (µM). 

#### 5.4.3. Confirmation of 5, 6,11-trideoxyTTX and Labeled Toxin by LC-qTOFMS

5,6,11-trideoxyTTX and labeled toxin were detected by a LC-QTOFMS (Triple TOF^TM^ 5600^+^; SCIEX, Framingham, MA, USA) as follows: Column, Atlantis HILIC (Waters, Milford, MA, USA), 3 µm, 2.1 × 150 mm; mobile phase A, acetonitrile; mobile phase B, 10 mM ammonium formate buffer (pH 4.0); flow rate, 0.2 mL/min; injection volume, 5 µL; CE values, 30 ± 10 eV; gradient, 0 min (B: 20%)—15 min (B: 60%)—16 min (B: 60%)—16.1 min (B: 20%)—19 min (B: 20%); mode, TOF-MS, operated in a positive ion mode via electrospray ionization.

#### 5.4.4. Inoculation of the Haptenic Antigen to Rabbits

The conjugate of KLH-TTX was immunized to five rabbits. We ordered the inoculation to rabbits and preparation of their sera to Protein Purify Co. (Isezaki, Gunma, Japan). In the first month, 1 mL of the antigen solution [0.3 mg/mL PBS(−)] was inoculated to one rabbit with complete adjuvant biweekly. After one month, rabbits were continuously immunized each with 1 mL solution of the antigen mixed with an incomplete adjuvant. The rabbits were inoculated 14 times in total for seven months with the antigen, and their sera were collected by exsanguination.

#### 5.4.5. Monitoring the Antibody Activity

Five mL of blood was collected from each rabbit just before immunization, and antibody activity against TTX was monitored after inoculation of the antigen. A 100 µL portion of rabbit sera were mixed each with 100 µL of TTX solutions corresponding to 2, 5, 10, 20, or 25 µM diluted with PBS(−), and allowed to stand for 30 min at room temperature. The mixtures were filtered through Nanosep 10 K Omega (Pall Corporation, Ann Arbor, MI, USA), and TTX in the filtrate was quantified by an HPLC fluorometric analyzer. The titer (antibody activity) was evaluated by the amount of TTX (nmol) trapped by the 1 mL antiserum. The mixture, in which the antiserum was replaced with PBS(−), was analyzed in the same way and used as a control. In addition, 100 µL portion each of 5,6,11-trideoxyTTX, 4-*epi*TTX, 4,9-anhydroTTX, and 11-oxoTTX were mixed with the sera collected from the rabbits at 7th inoculation, and treated with the same manner as described above to estimate the cross-reactivity.

### 5.5. Preparation of Biotin-TTX as a Competitive Labeled Toxin for ELISA

Biotin-labeled TTX (Biotin-TTX), a competitive labeled toxin for ELISA, was prepared as follows. Freeze-dried 4,9-anhydroTTX (5 µmol), derived from isolated TTX and purified on a Bio-Rex 70 column, was added with 310 mg (2 mmol) of (±)-dithiothreitol (DTT) in 3 mL of 0.05 M potassium phosphate buffer (pH 8.0) and the mixture incubated at 37 °C for 30 min. The adduct of TTX and DTT produced in the reaction mixture was isolated by column chromatography of Bio-Gel P-2. An aliquot of freeze-dried DTT-TTX (3.4 mol) was dissolved in 2 mL of 0.1 M sodium phosphate buffer (pH 7.4) and mixed with 4 mg of EZ-Link Maleimide-PEG2-Biotin (Thermo Fisher Scientific, Waltham, MA, USA). The mixture was allowed to stand for two hours at room temperature, loaded on a column of Bio-Gel P-2, eluted with water, and then with 0.2 M acetic acid. Biotin-TTX in the fractions eluted with 0.2 M acetic acid was monitored by the same method described in Section 5.4.2. The fractions which showed fluorescence peaks beyond 20 µM TTX equivalent were combined. The biotin-TTX in the combined solution was confirmed by high-resolution mass (Triple TOF^TM^ 5600^+^; SCIEX, Framingham, MA, USA) as follows: Mobile phase, water; flow rate, 0.2 mL/min; injection volume, 5 µL; CE values, 30 ± 10 eV; mode, TOF-MS, operated in a positive ion mode via electrospray ionization scanned from *m*/*z* 100 to 1000. The concentration of Biotin-TTX was determined with fluorometric analyzer as described above (Section 5.4.2). The test solutions were filtered through Nanosep 10K Omega prior to these analyses.

### 5.6. Preparation of Polyclonal Antibody

#### 5.6.1. Preparation of TTX-coupled Affinity Resin

EAH-Sepharose 4B (GE Healthcare) was washed with water at room temperature, and 10 mL portion of the resin was mixed with 10 mL PBS(−) and 56 mg GMBS dissolved in 2.5 mL DMSO. After 20 min, the slurry was poured into a glass column (2 × 3 cm) and washed with 200 mL water and with 50 mL PBS(−), successively. Freeze-dried 14 µmol DTT-TTX was dissolved in 30 mL PBS(−) and mixed with the GMBS-treated EAH-Sepeharose 4B. The mixed slurry was gently stirred at room temperature for one hour, and the column was washed with 100 mL water. To quench the remaining maleimide moiety on the resin, the column was washed with the mixture of 1 mL 2-mercaptoethanol and 100 mL PBS(−). The column was once again washed with 100 mL water containing 0.1 g sodium azide.

#### 5.6.2. Purification of Polyclonal Antibody

An aliquot of TTX-coupled affinity resin obtained as above was packed in a glass column (1 × 3 cm) and washed with 100 mL PBS(−). The precipitate obtained by 50% saturated ammonium sulfate fractionation from 7.5 mL portion of antiserum (Rabbit No.3, exsanguinated 2 weeks after the final inoculation) was dissolved in 7.5 mL PBS(−), loaded on the affinity column, and treated with 60 mL PBS(−) and 100 mL of 0.1 M glycine-HCl buffer (pH 2.7), successively, at room temperature. Each 2 mL fraction eluted with 0.1 M glycine-HCl buffer was collected in test tubes added beforehand with 400 L of 1 M 2-amino-2-hydroxymethyl-1,3-propanediol (Tris), while monitoring the absorption at 280 nm using a spectrophotometer (V-550, Jasco). Isolated antibody specific for TTX and its analogs thus obtained was mixed with an equivalent volume of PBS(−) containing 6 L ProClin 300 (Sigma-Aldrich, 48912-U), divided to each 1 mL, and stored in a deep freezer (−80 °C) until use.

#### 5.6.3. Direct One Step ELISA for TTX and Its Analogs

The isolated polyclonal antibody was diluted 100 times with 0.9% (*w*/*v*) NaCl in 0.01 M Tris-HCl buffer (pH 8.2), and 100 µL portions were added to each well of a 96-well ELISA plate (Maxisorp, Thermo Fisher Scientific). After stirring at 5 °C overnight, the plate was washed with PBS(−) and added each 350 L Block Ace (4% *w/v* in water; KAC Co., Ltd., Kyoto, Japan) solution and left at 5 °C overnight. The plate was washed with PBS(−) containing 0.05% (*v/v*) Tween 20 (PBST) twice. The assay procedure is summarized as follows: 50 L solution of isolated TTX or its analogs (11-oxoTTX, 4-*epi*TTX, 4,9-anhydroTTX, 5,6,11-trideoxyTTX) at 1–1000 nM in 0.1 M sodium phosphate buffer, pH 7.4, was added to each well of the ELISA plate coated with the antibody. Then, 50 L Biotin-TTX (2 nM, diluted with the same buffer) was added to each well and the plate was incubated at 37 °C for 15 min. While the solution was discarded, the wells were washed with PBST and added with 50 L HRP-Streptavidin solution [2000 times dilution with PBS(−); Funakoshi, Tokyo, Japan). The plate was incubated at 37 °C for 15 min and the solution was discarded. After washing the wells three times with PBST, 100 µL Sigma-Fast tablets (OPD-H_2_O_2_, Sigma-Aldrich, dissolved in 10 mL water) solution were added to each well, the plate was incubated at 37 °C for 5 min, and the wells were added with 100 µL HCl (2 M). Finally, each well was subjected to measurement of absorbance at 490 nm with a plate reader (iMark, Bio-Rad). The toxin solution samples were analyzed in triplicate.

### 5.7. Cross-Reactivity with Paralytic Shellfish Toxins

Paralytic shellfish toxins (PSTs) such as gonyautoxin 1/4 (GTX1+4, equilibrated mixture), GTX 2/3, C1/2, neosaxitoxin (neoSTX) and decarbamoylsaxitoxin (dcSTX) were isolated from contaminated bivalves according to the method described in Section 5.2. The concentrations of these toxins in the solution were confirmed by HPLC fluorometric analyzer developed by Oshima [41] in which toxin references were kindly supplied from Dr. Oshima. The isolated toxins were examined for their cross-reactivity on ELISA with the TTX antibody. Briefly, these toxins were dissolved separately in 0.1 M sodium phosphate buffer (pH 7.2) at 1–1000 nM and examined for their cross-reactivity in the same manner as described above. 

### 5.8. Ethics Statement

All animal experiments presented in this study were approved by Animal Experiments Ethical Review Board in Protein Purify Co: Jikken-Dobutsu-Rinri-Iinkai (K17A0710). Approval date: 10 July 2017.

## Figures and Tables

**Figure 1 toxins-11-00551-f001:**
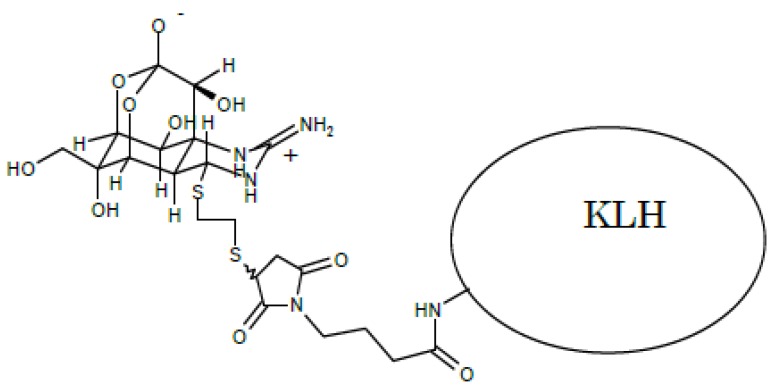
The estimated structure of the conjugated tetrodotoxin (TTX) antigen (KLH-GMBS-EDT-TTX).

**Figure 2 toxins-11-00551-f002:**
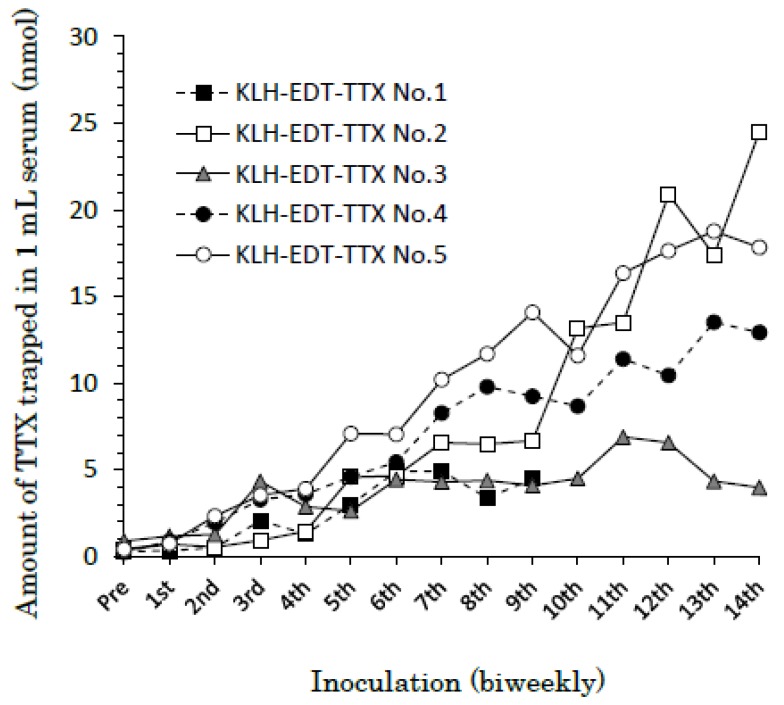
Changes in the titer of rabbits inoculated with KLH-TTX. Titers are expressed in the ordinate as the amount of TTX (nmol) adsorbed in 1 mL serum.

**Figure 3 toxins-11-00551-f003:**
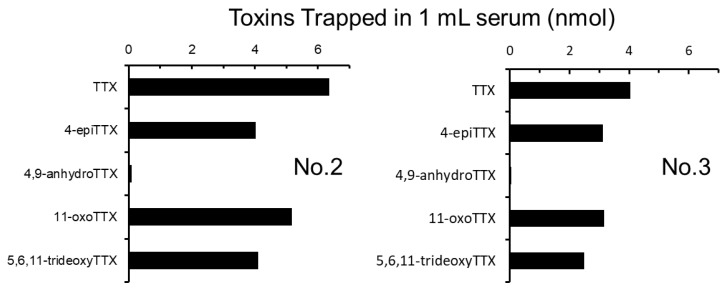
The amounts of TTX and its analogs trapped in a rabbit (No.2 and No.3) serum.

**Figure 4 toxins-11-00551-f004:**
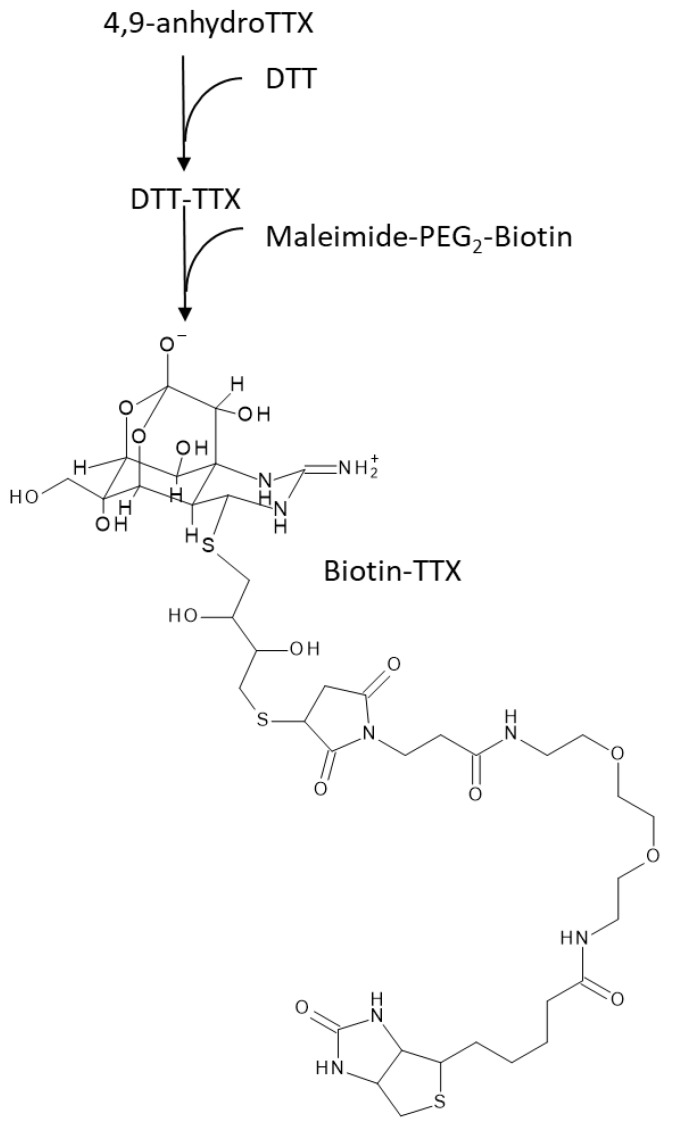
The estimation structure of the antigen labeled with biotin (Biotin-DTT-TTX).

**Figure 5 toxins-11-00551-f005:**
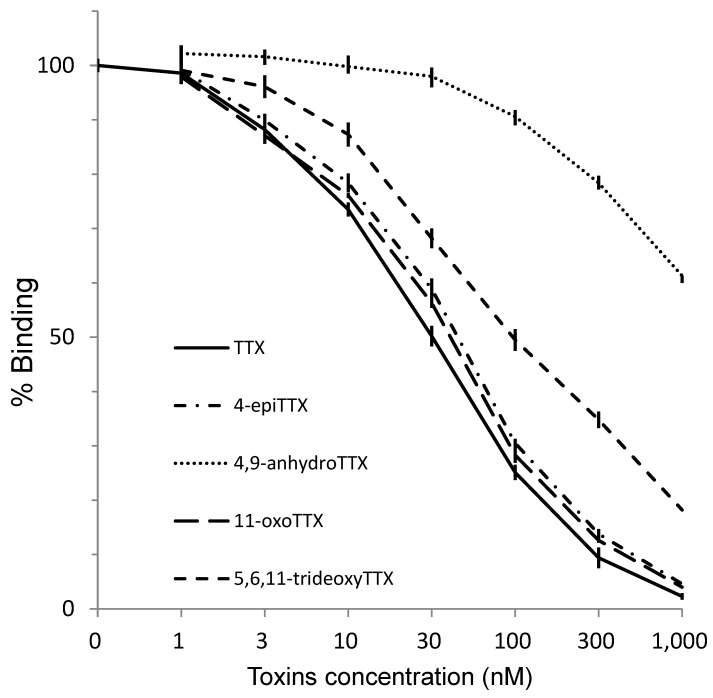
The reactivity of the antibody against TTX and its analogs in enzyme-linked immunosorbent assay (ELISA) (mean ± S.E., *n* = 3).

**Figure 6 toxins-11-00551-f006:**
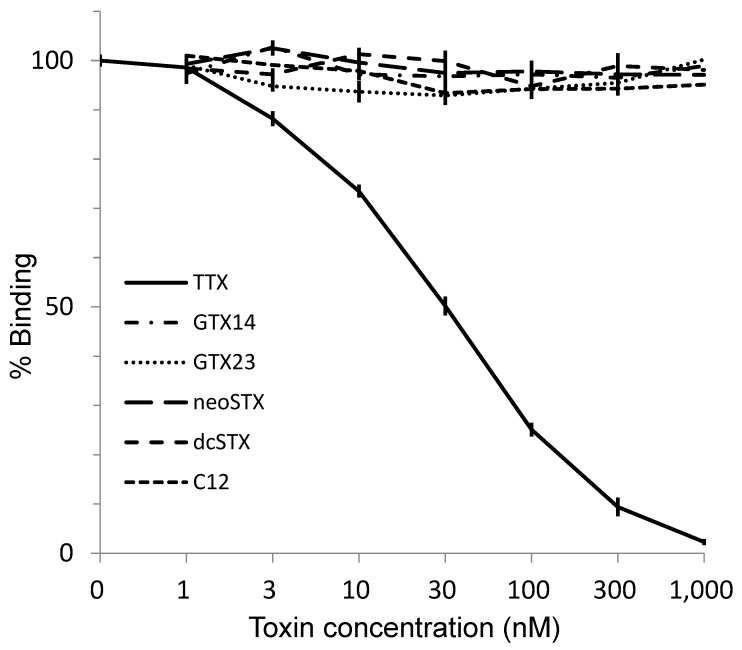
The reactivity of the antibody against paralytic shellfish toxins in ELISA (mean ± S.E., *n* = 3).

**Figure 7 toxins-11-00551-f007:**
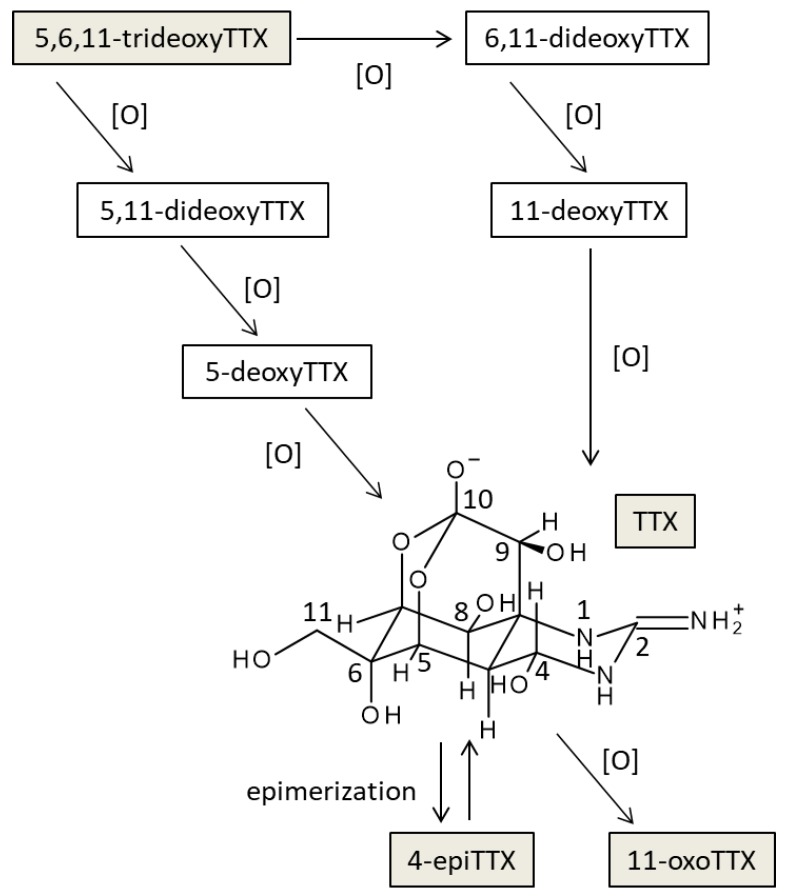
The estimated conversion pathways of TTX and its analogs in toxic organisms (partially modified from Yotsu-Yamashita et al. [19]).

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
