# Peer review of "Novel Polyclonal Antibody Raised against Tetrodotoxin Using Its Haptenic Antigen Prepared from 4,9-anhydrotetrodotoxin Reacted with 1,2-Ethaneditiol and Further Reacted with Keyhole Limpet Hemocyanin"

_toxins, 2019, doi:10.3390/toxins11100551_

Round 1

Reviewer 1 Report

I am very glad to see the interest in the quantification of TTXs by means of ELISA with polyclonal antibodies. The manuscript under reviewing provides development of novel polyclonal antibody against TTX and utilizing it direct one-step ELISA system for TTX, 4-epiTTX, 11-oxoTTX, 5,6,11-trideoxyTTX detection. While methods and results presented in the manuscript are adequate and appropriate, since the new method is provided, several additional validation studies are needed. My objections follow below:

Introduction section

Line 23-24. “vertebrates and invertebrates [8,9,10]”. All these references are connected only with vertebrates. Please, set references on TTX-bearing invertebrate animals.

Line 25. Noguchi et al (2008) was the first, who hypothesized TTX bioaccumulation through small zooplankton and detritus feeders along the food chain (see full link below). It would be better to include this reference here.

“Noguchi, T.; Arakawa, O. Tetrodotoxin—Distribution and accumulation in aquatic organisms, and cases of human intoxication. Mar. Drugs 2008, 6, 220–242.”

Line 27-30. Please, pay attention on two last works devoted to the TTXs toxins measurement in ribbon worms. Rephrase this sentence according to these works (doi.org/10.1016/j.toxicon.2018.11.006, doi:10.3390/md16110452).

Line 44. Reference 26. This work does not reveal fine localization of TTX. Only two articles by now are devoted to the TTX distribution on electron and light microscopy level (10.1016/S0041-0101(03)00003-5; 10.1016/j.toxicon.2016.01.060). In the case you mean “fine localization of TTX in TTX-containing animals” it is only light optical level you should pay attention to another articles (see below).

Campbell, M.E., Schwartz, M.L., 2008. Immunohistological visualization of tetrodotoxin in Micrura verrili and Dushia atra (Phylum Nemertea). In: National Conferences for Undergraduate Research (NCUR) (Proceedings). Loustalet, M., Campbell, M.B., Schwartz, M.L., 2009. Microdistribution of tetrodotoxin in three species of nemerteans. In: 7th International Conference on Nemertean Biology (Proceedings). Mahmud, Y., Okadaa, K., Takatani, T., Kawatsu, K., Hamano, Y., Arakawa, O., Noguchi, T., 2003. Intra-tissue distribution of tetrodotoxin in two marine puffers Takifugu vermicularis and Chelonodon patoca. Toxicon 41, 13-18. Mebs, D., Arakawa, O., Yotsu-Yamashita, M., 2010. Tissue distribution of tetrodotoxin in the red-spotted newt Notophthalmus viridescens. Toxicon 55, 1353-1357. Miyazawa, K., Jeon, J.K., Noguchi, T., Ito, K., Hashimoto, K., 1987. Distribution of tetrodotoxin in the tissues of the flatworm Planocera multitentaculata (Platyhelmithes). Toxicon 25, 975-980. Tanu, M.B., Mahmud, Y., Takatani, T., Kawatsu, K., Hamano, Y., Arakawa, O., Noguchi, T., 2002. Localization of tetrodotoxin in the skin of a brackishwater puffer Tetraodon steindachneri on the basis of immunohistological study. Toxicon 40, 103-106. Tanu, M.B., Mahmud, Y., Arakawa, O., Takatani, T., Kajihara, H., Kawatsu, K., Hamano, Y., Asakawa, M., Miyazawa, K., Noguchi, Y., 2004. Immunoenzymatic visualization of tetrodotoxin (TTX) in cephalothrix species (Nemertea: Anopla: Palaeonemertea: Cephalotrichidae) and Planocera reticulata (Platyhelminthes: Turbellaria: Polycladida: Planoceridae). Toxicon 44, 515-520. Williams, B.L., Stark, M.R., Caldwell, R.L., 2009. Intra-organismal distribution of tetrodotoxin in two species of blue-ringed octopuses (Hapalochlaena fasciata and H. lunulata). Toxicon 54, 345-353.

Line 52. “on molecular species of TTX derivatives” – molecular species seems to be unclear definition, please rephrase this part.

Result section

HPLS fluorometric analysis data is absent. It is desirable to include this data in results.

Figure 5. Please, specify Ox and Oy.

Figure 6. In my opinion it is better to change “% Binding” to the “Absorbance”. In this case difference between minimal and maximal values will be clearer and will allow to evaluate the sensitivity of the method.

Discussion section.

Line 126-129. Eangoor with colleagues developed ELISA technique for detection of summed PSTs concentration (DOI: 10.1093/jat/bkx072). It will be interesting to discuss this data in connection with your results here.

Materials and Methods section

The manuscript doesn’t provide any evaluation studies of ELISA technique. Data of the limits of detection, limits of quantification, working range, mean, standard deviation, recovery and coefficient of variation of the method is necessary (for example, https://www.ncbi.nlm.nih.gov/pmc/articles/PMC4541289/)).

Since KLH is marine organism’s protein, polyclonal antibodies that was obtained using this protein could cross-react with KHL which could be a component of assumed matrix (toxic aquatic organisms) despite antibodies affine purification. So, if authors have data on the cross-reactivity of their antibodies with KLH it would be better to present them.

Author Response

19.8.23/9.6

reviewer #1

I am very glad to see the interest in the quantification of TTXs by means of ELISA with polyclonal antibodies. The manuscript under reviewing provides development of novel polyclonal antibody against TTX and utilizing it direct one-step ELISA system for TTX, 4-epiTTX, 11-oxoTTX, 5,6,11-trideoxyTTX detection. While methods and results presented in the manuscript are adequate and appropriate, since the new method is provided, several additional validation studies are needed. My objections follow below:

Introduction section

Line 23-24. “vertebrates and invertebrates [8,9,10]”. All these references are connected only with vertebrates. Please, set references on TTX-bearing invertebrate animals.

We added a review (Biessy et al. 2019) as reference [11] for “invertebrates”.

Line 25. Noguchi et al (2008) was the first, who hypothesized TTX bioaccumulation through small zooplankton and detritus feeders along the food chain (see full link below). It would be better to include this reference here

“Noguchi, T.; Arakawa, O. Tetrodotoxin—Distribution and accumulation in aquatic organisms, and cases of human intoxication. Mar. Drugs 2008, 6, 220–242.”

We added this as reference [17].

Line 27-30. Please, pay attention on two last works devoted to the TTXs toxins measurement in ribbon worms. Rephrase this sentence according to these works (doi.org/10.1016/j.toxicon.2018.11.006, doi:10.3390/md16110452).

We added reference [24] for the occurrence of TTXs in ribbon worms.

Line 44. Reference 26. This work does not reveal fine localization of TTX. Only two articles by now are devoted to the TTX distribution on electron and light microscopy level (10.1016/S0041-0101(03)00003-5; 10.1016/j.toxicon.2016.01.060). In the case you mean “fine localization of TTX in TTX-containing animals” it is only light optical level you should pay attention to another articles (see below).

Campbell, M.E., Schwartz, M.L., 2008. Immunohistological visualization of tetrodotoxin in Micrura verrili and Dushia atra (Phylum Nemertea). In: National Conferences for Undergraduate Research (NCUR) (Proceedings). Loustalet, M., Campbell, M.B., Schwartz, M.L., 2009. Microdistribution of tetrodotoxin in three species of nemerteans. In: 7th International Conference on Nemertean Biology (Proceedings). Mahmud, Y., Okadaa, K., Takatani, T., Kawatsu, K., Hamano, Y., Arakawa, O., Noguchi, T., 2003. Intra-tissue distribution of tetrodotoxin in two marine puffers Takifugu vermicularis and Chelonodon patoca. Toxicon 41, 13-18. Mebs, D., Arakawa, O., Yotsu-Yamashita, M., 2010. Tissue distribution of tetrodotoxin in the red-spotted newt Notophthalmus viridescens. Toxicon 55, 1353-1357. Miyazawa, K., Jeon, J.K., Noguchi, T., Ito, K., Hashimoto, K., 1987. Distribution of tetrodotoxin in the tissues of the flatworm Planocera multitentaculata (Platyhelmithes). Toxicon 25, 975-980. Tanu, M.B., Mahmud, Y., Takatani, T., Kawatsu, K., Hamano, Y., Arakawa, O., Noguchi, T., 2002. Localization of tetrodotoxin in the skin of a brackishwater puffer Tetraodon steindachneri on the basis of immunohistological study. Toxicon 40, 103-106. Tanu, M.B., Mahmud, Y., Arakawa, O., Takatani, T., Kajihara, H., Kawatsu, K., Hamano, Y., Asakawa, M., Miyazawa, K., Noguchi, Y., 2004. Immunoenzymatic visualization of tetrodotoxin (TTX) in cephalothrix species (Nemertea: Anopla: Palaeonemertea: Cephalotrichidae) and Planocera reticulata (Platyhelminthes: Turbellaria: Polycladida: Planoceridae). Toxicon 44, 515-520. Williams, B.L., Stark, M.R., Caldwell, R.L., 2009. Intra-organismal distribution of tetrodotoxin in two species of blue-ringed octopuses (Hapalochlaena fasciata and H. lunulata). Toxicon 54, 345-353.

We removed “Kodama et al.[26]” from the text, and added 4 references: Mahmud et al. [34], Mebs et al.[31], Williams et al.[30], and Miyazawa et al.[29] for localization of TTX in tissues of toxic animals.

Line 52. “on molecular species of TTX derivatives” – molecular species seems to be unclear definition, please rephrase this part.

We changed “molecular species of TTX derivatives ” to “toxin structures”.

Result section

HPLS fluorometric analysis data is absent. It is desirable to include this data in results.

Figure 5. Please, specify Ox and Oy.

We added “Toxin concentration (nM)” for x-axis and “% Binding” for y-axis.

Figure 6. In my opinion it is better to change “% Binding” to the “Absorbance”. In this case difference between minimal and maximal values will be clearer and will allow to evaluate the sensitivity of the method.

TTX and 5 PST components with different concentrations (0, 1,3,10,30,100,300, and 1000 nM) were analyzed on 96 well plates in triplicate. In Figure 5, five TTX analogues were analyzed by using three plates. The absorbance is different between the plates as shown below. We use “% Binding” to compare different TTX analogues determined with different plates in one figure.

Discussion section.

Line 126-129. Eangoor with colleagues developed ELISA technique for detection of summed PSTs concentration (DOI: 10.1093/jat/bkx072). It will be interesting to discuss this data in connection with your results here.

We added a short discussion on the sensitivity of our ELISA for TTXs, with two references reporting ELISA for PSTs (Eangoor et al. 2017 [39] and Sato et al. 2014 [36]).

Materials and Methods section

The manuscript doesn’t provide any evaluation studies of ELISA technique. Data of the limits of detection, limits of quantification, working range, mean, standard deviation, recovery and coefficient of variation of the method is necessary (for example, https://www.ncbi.nlm.nih.gov/pmc/articles/PMC4541289/)).

We added the detection limit (3 nM) and working range (10-100 nM) for TTX on newly developed ELISA in Results section (2.5).

Since KLH is marine organism’s protein, polyclonal antibodies that was obtained using this protein could cross-react with KHL which could be a component of assumed matrix (toxic aquatic organisms) despite antibodies affine purification. So, if authors have data on the cross-reactivity of their antibodies with KLH it would be better to present them.

Thank you very much. According the comments, we examined KLH on the same ELISA plate with TTX in triplicate, and obtained the results shown below (KLH did not react with the antibody). We added this description in Results section (2.5).

Table. The reactivity of the antibody against KLH in ELISA

OD (n=3)

OD (n=3)

TTX (nM)

mean

SE

KLH (ppm)

mean

SE

0

0.702

0.023

0

0.702

0.017

1

0.698

0.019

0.1

0.701

0.021

10

0.423

0.023

1

0.67

0.022

100

0.225

0.023

10

0.67

0.021

1000

0.151

0.023

100

0.607

0.019

Bound 0 %

0.124

0.021

Bound 0 %

0.124

0.017

TTX and KLH with several concentration were separately analyzed on the same ELISA plate. Back ground OD (absence of biotin-TTX) is expressed as “Bound 0%.

Reviewer 2 Report

The manuscript describes the development of a novel polyclonal antibody against TTX using direct one step ELISA. Since the TTX is a potent toxin it is of outmost interest to have a sensitive method for its detection. However, not only TTX but also its analogues should be included in a detection technique.

TTX analogues such as 4-epiTTX, 11-oxoTTX, 5,6,11-trideoxyTTX showed high reactivity to the antibody in ELISA, whereas 4,9-anhydroTTX did not. Additionally no cross-reaction was observed for PST components. This is a good characteristics since the test may distinguish between PST and TTX toxins and point out towards further confirmations.

The manuscript may benefit from improving the consistency and link between the methods and results section. Moreover, there are multitude data that are produced in the study but not reported. This is mostly related to the toxins used in the study, but also to other materials. The manuscript should be refined and the discussion of the results should offer the clear added value of developed antibody and the prespectives of their use.

Herebelow the authors may find more specific comments:

Figure 2: Direct one step ELISA was carried out using the purified antibody obtained from rabbit No.3. but the absorption of TTX and analogues by rabbit No 2 was shown.

Figure 3: it appears that 4,9 anhydro-TTX was hardly absorbed in serum. Could it be that his had impact on some of your results? What is the role of intra-species variety here and was that observed and how was it handled in the study?

Figure 5: add the title of the y and x-axis

Discussion section (line 112-129): the authors are giving the conclusion of the study. The real discussion of the results obtained by the newly developed polyclonal antibodies in the view of already existent detection methods is not performed. While in the introduction there was some reasoning on the necessary of the potent assay and the lack of standard for MS analyses, The discussion is vague and indicates the possible conversion pathways. This is to certify why the prevalence of TTX and its analogues is possible in some organisms. The authors have to calrify this paragraph.

4.1 materials and reagents: whereas fish material used for extraction of TTX is described, there is no description of the reagents used in the study. This has to be added.

In the part, the authors wrote “alternatively, these were supplied from…” Why there was a need to search for an alternative and in which conditions the alternative supply was requested? This has to be amended to describe the sampling with necessary info.

4.2 Preparation of TTX and its analogues – hereby the extraction protocols were briefly explained. As a results TTX and its four analogues were produced. The total volume as well as the extraction yield could be interesting but due to the high potency of the TTX and possible misuse (security issues) it is not necessary to use exact numbers and also due to the fact that the protocol exists. On the other hand the purity of these compounds has to be verified . How was this done? Or was this part of the fluorimetric HPLC or TOF-MS analysis? Please respond or add necessary info to ensure the quality of obtained compounds.

4.4 Analytical procedures - this section describes the procedures but the results and data procured in this way are not clearly indicated in the results section. The authors are mentioning some results of TOF-MS analysis in the section 2.4. This, however lacks the clear reasoning, and the linkage between materials and results section is broken.

4.4.1 This is a quantitative fluorimetric method. This in turn means that quantification data of TTX, 4-epi-TTX and 4,9-anhydro TTX should have been obtained. Can authors provide this in the results? Furthermore, was there any correlation done with the reactivity and quantified data? At which wavelength was absorbance detector set?

4.4.2: what does it mean: “as shown in i)” (line 182); Here it was stipulated that the fluorimetric peak intensity was compare to that of TTX. Does this mean that quantification was done relative to TTX, which means expressed as TTX equivalent? Since these results are not supplied it is not fully clear.

4.4.3   line 184: Hereby the LC-qTOF-MS analysis is described. The instrument specification is lacking (important for a sensitivity deduction and evaluation), units are missing (line 188 for CE). MRM mode was used but what were the targets, describe them with this MS characteristics.

4.4.4 in total 5 rabbits were used – can you indicate the variety and gender (line 192). Hereby “continuously” (line 194) was used to explained immunization. Over which period the rabbits received 1ml solution and how many immunization were there finally? It appears from line 205 that there were 7 immunizations. Is that correct? This part has to be amended.

4.4.5 TTX was quantified in the filtrate (line 201). The authors need to be more specific for which purposed quantification was used and how this quantified data were expressed. An additional table of supplementary material may be considered.

4.5 preparation of Biotin-TTX: in line 209 the authors say that 4,9 anhydroTTX was derived from isolated TTX. Is that according to the procedure described in 4.2? if yes, it is necessary to specify it.

Line 216: monitoring of biotin-TTX means that this was only a qualitative analysis to confirm the formation of the complex. Was there any threshold or detection limit established for this procedure? This monitoring was done in gel after water ad acetic acid elution. Was there any other preparation prior to LC-qTOF-MS analysis performed (eg. Filration )?

4.6.2 purification: it is described that each 2ml fraction (line 236-237) was collected and analysed by observing the absorption at 280nm. Does this mean 2ml of collected was (60ml plus 100ml) or from the volume 7,5+7,5 ml? Finally, what kind of instrument was used for observing the absorption and which fractions were taken?

4.7 PST cross-reactivity: it appears that the authors worked with isolated Paralytic shellfish toxins. It is necessary to describe which procedure was used for this and how were the toxins identified? Did authors used any certified reference standards for the toxins? For any quantification the use of certified standards is recommended.

Author Response

19.8.23

The manuscript describes the development of a novel polyclonal antibody against TTX using direct one step ELISA. Since the TTX is a potent toxin it is of outmost interest to have a sensitive method for its detection. However, not only TTX but also its analogues should be included in a detection technique.

TTX analogues such as 4-epiTTX, 11-oxoTTX, 5,6,11-trideoxyTTX showed high reactivity to the antibody in ELISA, whereas 4,9-anhydroTTX did not. Additionally no cross-reaction was observed for PST components. This is a good characteristics since the test may distinguish between PST and TTX toxins and point out towards further confirmations.

The manuscript may benefit from improving the consistency and link between the methods and results section. Moreover, there are multitude data that are produced in the study but not reported. This is mostly related to the toxins used in the study, but also to other materials. The manuscript should be refined and the discussion of the results should offer the clear added value of developed antibody and the prespectives of their use.

Here below the authors may find more specific comments:

Figure 2: Direct one step ELISA was carried out using the purified antibody obtained from rabbit No.3. but the absorption of TTX and analogues by rabbit No 2 was shown.

We added the results of rabbit No.3 in Figure 3.

Figure 3: it appears that 4,9 anhydro-TTX was hardly absorbed in serum. Could it be that his had impact on some of your results? What is the role of intra-species variety here and was that observed and how was it handled in the study?

We added discussions about the low cross-reactivity of 4 ,9-anhTTX in Discussion section (Line 128 ~).

Figure 5: add the title of the y and x-axis

We added the titles in Fig.5.

Discussion section (line 112-129): the authors are giving the conclusion of the study. The real discussion of the results obtained by the newly developed polyclonal antibodies in the view of already existent detection methods is not performed. While in the introduction there was some reasoning on the necessary of the potent assay and the lack of standard for MS analyses, The discussion is vague and indicates the possible conversion pathways. This is to certify why the prevalence of TTX and its analogues is possible in some organisms. The authors have to calrify this paragraph.

We added the related discussions in page 6 (Line 128~145)

4.1 materials and reagents: whereas fish material used for extraction of TTX is described, there is no description of the reagents used in the study. This has to be added.

“Reagents” was removed from the title.

In the part, the authors wrote “alternatively, these were supplied from…” Why there was a need to search for an alternative and in which conditions the alternative supply was requested? This has to be amended to describe the sampling with necessary info.

We changed this sentence to “Frozen samples of T. rubripes and L. scerelatus were supplied from ---“ as shown in section 4.1.

4.2 Preparation of TTX and its analogues – hereby the extraction protocols were briefly explained. As a results TTX and its four analogues were produced. The total volume as well as the extraction yield could be interesting but due to the high potency of the TTX and possible misuse (security issues) it is not necessary to use exact numbers and also due to the fact that the protocol exists. On the other hand the purity of these compounds has to be verified . How was this done? Or was this part of the fluorimetric HPLC or TOF-MS analysis? Please respond or add necessary info to ensure the quality of obtained compounds.

We provided the fluorometric HPLC data (chromatograms) as a supplementary material.

Supplementary Figure 1. Fluorometric HPLC analysis of isolated TTX analogues.

4.4 Analytical procedures - this section describes the procedures but the results and data procured in this way are not clearly indicated in the results section. The authors are mentioning some results of TOF-MS analysis in the section 2.4. This, however lacks the clear reasoning, and the linkage between materials and results section is broken.

4.4.1 This is a quantitative fluorimetric method. This in turn means that quantification data of TTX, 4-epi-TTX and 4,9-anhydro TTX should have been obtained. Can authors provide this in the results? Furthermore, was there any correlation done with the reactivity and quantified data? At which wavelength was absorbance detector set?

Details of HPLC analysis, such as wavelength (Ex, Em) is shown in section 4.4.1. We provided the data as supplementary material as shown above.

4.4.2: what does it mean: “as shown in i)” (line 182); Here it was stipulated that the fluorimetric peak intensity was compare to that of TTX. Does this mean that quantification was done relative to TTX, which means expressed as TTX equivalent? Since these results are not supplied it is not fully clear.

This is our mistake. As shown in the revised version, we corrected to “4.4.1”.

4.4.3   line 184: Hereby the LC-qTOF-MS analysis is described. The instrument specification is lacking (important for a sensitivity deduction and evaluation), units are missing (line 188 for CE). MRM mode was used but what were the targets, describe them with this MS characteristics.

According to the comments, we added the information.

4.4.4 in total 5 rabbits were used – can you indicate the variety and gender (line 192). Hereby “continuously” (line 194) was used to explained immunization. Over which period the rabbits received 1ml solution and how many immunization were there finally? It appears from line 205 that there were 7 immunizations. Is that correct? This part has to be amended.

As newly described in this section, we ordered the immunization and collect sera to a company (Protein Purity Co.). We do not have information about gender of rabbits.

4.4.5 TTX was quantified in the filtrate (line 201). The authors need to be more specific for which purposed quantification was used and how this quantified data were expressed. An additional table of supplementary material may be considered.

We added phrases with “ “ in the text to define clearer quantification as follows. The titer (antibody activity) was evaluated by the amount of TTX “(nmol)” trapped by the “1 mL” antiserum. We think the additional table is not necessary, since the data is shown in Figure 2.

4.5 preparation of Biotin-TTX: in line 209 the authors say that 4,9 anhydroTTX was derived from isolated TTX. Is that according to the procedure described in 4.2? if yes, it is necessary to specify it.

--- derived from isolated TTX (and purified on a Bio-Rex 70 column), ---. We added the sentence with () in the text.

Line 216: monitoring of biotin-TTX means that this was only a qualitative analysis to confirm the formation of the complex. Was there any threshold or detection limit established for this procedure? This monitoring was done in gel after water ad acetic acid elution. Was there any other preparation prior to LC-qTOF-MS analysis performed (eg. Filration )?

Biotin-TTX was detected by TOF-MS, not by LC-qTOF-MS. We specified the analytical conditions in the text.

4.6.2 purification: it is described that each 2ml fraction (line 236-237) was collected and analysed by observing the absorption at 280nm. Does this mean 2ml of collected was (60ml plus 100ml) or from the volume 7,5+7,5 ml? Finally, what kind of instrument was used for observing the absorption and which fractions were taken?

We changed the original description to that with underlines as follows. Each 2 mL fraction eluted with 0.1 M glycine-HCl buffer was collected in test tubes added beforehand with 400 mL of 1 M 2-amino-2-hydroxymethyl-1,3-propanediol (Tris), while monitoring the absorption at 280 nm using a spectrophotometer (V-550, Jasco).

4.7 PST cross-reactivity: it appears that the authors worked with isolated Paralytic shellfish toxins. It is necessary to describe which procedure was used for this and how were the toxins identified? Did authors used any certified reference standards for the toxins? For any quantification the use of certified standards is recommended.

According to the comments, we added information requested.

Round 2

Reviewer 1 Report

My minor objections/suggestions are listed below:

Line 24. For invertebrates it will be good to use reference on the review. For example, DOI:10.3390/md20080011

Line 26. Yamada et al (2017) did not postulate “accumulation TTX through food webs”. Please remove this reference.

Line 27-30. Could you insert references for each statement. Now it is not clear in which works it is shown that “their concentrations in toxic organisms are much lower than that of TTX”, and in which – “except 5,6,11-trideoxyTTX”.

Line 30: Turner et al, 2018 stated “The parent TTX was the dominant analogue, corresponding to 64%”. So, it will be good to use other reference for this sentence – namely https://doi.org/10.1016/j.toxicon.2018.11.006. In this work authors revealed “The TTXs concentrations in the C. simula extract decrease as follows: 5,6,11-trideoxyTTX>5-deoxyTTX > TTX>11-deoxyTTX>4-epiTTX>4,9-anhydroTTX>11-norTTX-6(S)-ol > 11-norTTX-6(R)-ol.”

Line 102-103. Please add also limit of quantification.

“Material and Methods” section:

From the standard curve data it should be possible to calculate an association constant, which would be a valuable measure.

Author Response

We revised according to all of your comments.

Reviewer 2 Report

In this revision of the manuscript the authors point out clearly that this is a study where an antibody against TTX and it analogues is developed followed by direct application in an ELISA test. Among TTX analogues 4-epiTTX, 11-oxoTTX, 5,6,11-trideoxyTTX showed high reactivity to the antibody in ELISA, whereas 4,9-anhydroTTX did not. Furthermore, the message is that this antibody/ELISA is TTX specific. For this a lack of cross-reaction with PST toxins was confirmed in the study.

The authors discussed their results in the view of other reported results. The figures are improved. In this version the focus of the research is clear and the used resources are beter explained.

Regarding the figures the authors are invited to check that all annotations are done in the same manner (eg 4,9-anhydroTTX vs 4,9 anhydroTTX). The results obtained from rabbit No3 were added in Figure 3 since this the purified antibody from this mouse was used further in the study. Please adapt the sentence in line 80 to correspond this (The amounts of TTX and its analogues trapped in No.2 rabbit are shown in Figure 3. )

The authors are also invited to provide explanation on selecting the rabbit No3 over the rabbit No2 which shows apparently better results (Fig3). If this was a random selection it should be as well explained, but if it was based on the results it is necessary to clarify it.

4.4.1. TTX, 4-epiTTX, and 4,9-anhydroTTX: change title into Toxin confirmation by HPLC-FLD and then specify which toxins were confirmed by this technique

4.4.3. 5, 6,11-trideoxyTTX and Labeled Toxin: change the title into Toxin confirmation by LC-qTOFMS and then specify which toxins were confirmed by this technique

Line 156-157: The authors state the following: “Some improvements may be necessary on our ELISA system for TTX and its analogues”. What kind of imrpovements do the authors refer to? It is recommended to beter specify the limitation of the ELISA like 4,9-anhydroTTX is out of scope or low reactivity with this toxin, or detection limit is comparable or not comparable to other tests etc

Supplementary material:

It is very unusual to see HPLC chromatograms in vertical position. It could have been easier to see the full length chromatogram where you can indicate the retention time of each toxin with the intensity as a vertical y-axis. In case that the toxins were analysed in apart runs you may show them independently.

Modify the title of Supplemental Table 1 into more appropriate and more self-sufficient, eg: Reaction of TTX and its analogues with DDT expressed as concentration of the remaining free toxins measured by HPLC-FLD in function of time

Supplemental table 2: give full name for the abbreviation SE.

Author Response

(The authors gave the same response as above.)
